# In Silico Drug Design and Analysis of Dual Amyloid-Beta and Tau Protein-Aggregation Inhibitors for Alzheimer’s Disease Treatment

**DOI:** 10.3390/molecules28031388

**Published:** 2023-02-01

**Authors:** Nisha Job, Venkatesan S. Thimmakondu, Krishnan Thirumoorthy

**Affiliations:** 1Department of Chemistry, School of Advanced Sciences, Vellore Institute of Technology, Vellore 632014, Tamil Nadu, India; 2Department of Chemistry and Biochemistry, San Diego State University, San Diego, CA 92182, USA

**Keywords:** Alzheimer’s disease, tau protein, amyloid beta, dual inhibitors, molecular docking, MD simulation, TD-DFT

## Abstract

Alzheimer’s disease (AD) is a progressive and irreversible neurodegenerative disorder that gradually leads to the state of dementia. The main features of AD include the deposition of amyloid-beta peptides (Aβ), forming senile plaques, and the development of neurofibrillary tangles due to the accumulation of hyperphosphorylated Tau protein (p-tau) within the brain cells. In this report, seven dual-inhibitor molecules (L_1–7_) that can prevent the aggregation of both Aβ and p-tau are suggested. The drug-like features and identification of the target proteins are analyzed by the in silico method. L_1–7_ show positive results in both Blood–Brain Barrier (BBB) crossing and gastrointestinal absorption, rendering to the results of the permeation method. The molecular docking test performed for L_1–7_ shows binding energies in the range of −4.9 to −6.0 kcal/mol towards Aβ, and −4.6 to −5.6 kcal/mol for p-tau. The drug’s effectiveness under physiological conditions is assessed by the use of solvation models on the investigated systems. Further, the photophysical properties of L_1–3_ are predicted using TD-DFT studies.

## 1. Introduction

Alzheimer’s disease (AD) is a progressive and irreversible neurodegenerative disorder that slowly results in the state of dementia [1,2]. Several theories have been proposed to explain the root cause of AD, whilst the exact mechanism is still under debate [3,4]. The amyloid-β (Aβ) cascade hypothesis, cholinergic hypothesis, tau hypothesis, oxidative stress hypothesis, and inflammation hypothesis explain the possible causes of AD in different ways [5,6]. The main features of AD include the deposition of β-amyloid peptides forming senile plaques on the extracellular surface of neurons and the development of neurofibrillary tangles due to the accumulation of hyperphosphorylated Tau protein within the brain cells [7].

Senile plaques, which are made up of Aβ peptides, are found in AD patients. Aβ peptides are the proteolytic product of the amyloid precursor protein (APP). The proteolysis of the amyloid precursor protein produces Aβ-peptides. The splitting of APP in the N-terminus by β-secretase on the cell surface results in the generation of the soluble amyloid precursor protein β-fragment (APPs β). Additionally, the γ-secretase cleaves the C-terminus (C99), which is attached to the cell membrane, resulting in the formation of Aβ-peptides. The aggregation of these peptides causes the formation of insoluble fibrils, which eventually clump together to form the distinctive Aβ plaques as seen in AD patients. However, some other Aβ species mutations favor the longer peptides that terminate in residues 40 (Aβ40) and 42 (Aβ42). The main factor in the formation of senile plaques has been identified as Aβ42 because of its hydrophobic nature and the fact that it aggregates more quickly [8,9].

The p-tau protein is a microtubule-associated protein that is mostly found in the central nervous system (CNS). The functionality of the tau protein includes the stabilization of microtubules in the nerve cells. The alternative splicing of a specific gene called microtubule-associated protein tau (MAPT) produces tau proteins. Normal p-tau regulates gene transcription and cell-cycle activity, and is required for signaling molecules [10]. Under normal circumstances, tau is highly stable; however, different variables can trigger tau aggregation, resulting in neurofibrillary tangles [11]. The microtubule-binding region is known as the projection domain, which plays a vital role in structural stabilization and is generally flanked by tau-phosphosites. The hyperphosphorylation of p-tau causes the microtubule organization to be disrupted [12,13]. In the tau protein, around 80 phosphorylation sites for tyrosine (Tyr), serine (Ser), and threonine (Thr) exist. These sites contain a variety of Tyr, Ser, and Thr phosphatase residues that are involved in phosphorylation. Many kinases are responsible for the regulation of p-tau phosphorylation, including protein kinase A (PKA), protein kinase C (PKC), glycogen synthase kinase-3 (GSK3), and cyclin-dependent kinases (CDKs). In a healthy physiological state, there should be a balance between phosphate and kinase functions leading to normal tau behavior. Most of the hyperphosphorylation regions of p-tau are encoded within three sites (Tyr, Ser, and Thr) that also play a major role in AD progression [14,15]. Hence, the abnormalities of p-tau can be defined as a highly contributing factor to AD.

The current therapeutic strategies employed for the treatment of AD are centered on single-target medications, which are still unable to control the disease’s progression. Studies showed that both Aβ and p-tau influence the cognitive status by synergistically interfering with the neurological pathways. Multi-target drugs offer a single molecular entity to treat several AD-related variables, which has significantly grown in popularity as a potential AD cure [11]. We selected a p-tau inhibitor containing piperidine-substituted 3-nitrobenzaldehyde pharmacophore unit as a reference molecule from the literature, and tried out several combinations to make its derivatives multi-targeted for both Aβ and p-tau to improve their efficiency and to possess Blood–Brain Barrier (BBB) permeability [16]. We employed various in silico approaches, including molecular docking, molecular dynamics, and pharmacokinetic evaluations, for finding new lead molecules for AD treatment. These in silico methods can improve drug efficiency and reduce experimental costs as well. We have also considered only synthetically feasible molecules for conducting the in silico studies.

The first criterion for designing suitable drug molecules for AD treatment is to analyze their inhibitory potency towards Aβ and p-tau aggregation. The second criterion is BBB-crossing and gastrointestinal-absorption ability. Gastrointestinal absorption is vital in terms of drug delivery. For AD treatment, BBB permeability is essential to prevent the aggregation of different proteins inside the human brain. Herein, we designed multiple organic molecules as dual Aβ/p-tau aggregation inhibitors. Based on the available literature, the N-heterocycle substituted 3-nitrobenzaldehyde scaffold was chosen as the core structure. Since these molecules are achievable via different synthetic routes, their experimental analysis is also possible after theoretical confirmation. Each molecule was shortlisted based on target prediction and BBB permeability using SwissADME tools. The seven selected molecules shown in Figure 1 exhibit BBB crossing, higher Aβ, and p-tau binding affinity. We calculated the energies of frontier molecular orbitals (FMOs) in all the molecules to understand their global reactivity descriptors and molecular stability. The seven selected molecules’ 3D structures are depicted in Appendix A. The aggregation of the proteins Aβ/p-tau is widely considered the root cause of AD. These seven chosen molecules can all selectively interact with these proteins, helping in their aggregation inhibition. This inhibitory action can only be achieved when the molecules bind in the particular protein position, which avoids aggregation. The primary reason for protein aggregation is the interaction between peptide residues of the protein. To understand these binding mechanisms, we analyze the interaction probability of the designed molecules towards the proteins using molecular docking. The best docking poses are further subjected to molecular dynamics simulation to understand the stability of these interactions. Further, molecules are analyzed using solvation models to mimic physiological conditions. Then, the TD-DFT studies of these molecules are carried out to check their photophysical properties under physiological conditions.

## 2. Results and Discussion

### 2.1. Prediction of Drug-Target and ADMET Properties

ADMET profile and binding efficiency are the cornerstones of the drug discovery journey. It is paramount to obtain the desired results in these studies for drug candidate molecules before entering the experimental phase [17,18]. In this work, computational tools were used to determine the ADMET features of the candidate molecules. The results obtained from the SwissADME tool are listed in Table 1. The total polar surface area (TPSA) is an important factor in terms of drugs’ bioavailability. The compounds with TPSA > 140 are passively absorbed and have very limited oral bioavailability. The TPSA value of the designed inhibitor molecules in the present work is in the range of 60–80, which is optimal in terms of bioavailability. The log P and log S values are used to measure the lipophilicity and solubility of the molecules that are in the range of −0.4 to 5.6 and −10 to 6, respectively [19]. All seven compounds were within the expected range, which indicates that they have sufficient absorbency and solubility. Skin permeability was evaluated by log K_p_ values, and the highest skin permeability was shown by L_1_ among the seven molecules. Lipinski’s rule [18] and Ghose’s rule [20] were used for evaluating the drug-likeness properties of the compounds. The rules showed zero violations against promoting the compounds as potential drug candidates. The toxicity data of the seven inhibitor molecules are listed in Appendix A. All the molecules are under the class 4 category with median lethal dose (LD_50_) values between 300 and 2000.

The potential drug candidates were identified by the SwissTargetPrediction. The results showed the inhibitory nature of the compounds toward both Aβ as well as p-tau. Both proteins are responsible for the progression of AD. The gastrointestinal-absorption and BBB-penetration data obtained from the BOILED-Egg method are given in Figure 2. The molecules are represented inside the BOILED-Egg model as either blue, which are substrates of P-glycoprotein, or red, which are not. All seven compounds are located in the yellow region and represented as a red circle in the model, which indicates the BBB-crossing and human gastrointestinal-absorption ability of these molecules. It is also clear that these molecules are not a substrate of P-glycoprotein, which indicates that these compounds cannot be actively effluxed from the CNS.

### 2.2. Frontier Molecular Orbital and Global Reactivity Descriptors

According to the widely recognized frontier molecular orbital (FMO) theory, chemical reactions are primarily driven by the frontier orbitals, specifically the highest occupied molecular orbital (HOMO) and the lowest unoccupied molecular orbital (LUMO) [21]. It is considered a dependable resource for determining the reactivity of small organic molecules. The energy values, hardness, softness, electronegativity, chemical potential, electrophilicity index, and maximum charge transfer index are calculated using HOMO and LUMO [22] energy levels, and are included in Table 2.

The band gap between the HOMO and LUMO is predicted in the range of 6.9–7.6 eV, and all the molecules can be considered biocompatible as reported in the literature, since the band gap is greater than 2.5 eV [23]. L_7_ has the lowest band-gap value, highest chemical potential, and lowest hardness value. The low chemical softness (σ) and high electrophilicity index (ω) show low cytotoxicity and good biological activity [24]. The lowest σ value of L_3_ represents the lowest cytotoxicity of all the molecules, and the high ω value (3.3) of L_5_ shows the highest biological activity [24,25]. The maximum charge transfer index (ΔN_max_) is the capacity of an electrophile that can accept a maximum electronic charge from the outer environment and is in the range of 1.1–1.4 for all the molecules [24].

### 2.3. Molecular Docking Analysis

As per the target prediction results, Aβ and p-tau have the probability to interact with molecules L_1–7_. Molecular docking studies help to understand the interacting position, affinity, and mechanism of the inhibitor molecules with the target proteins. Two sequence segments of p-tau with six amino-acid residues, _275_VQIINK_280_ at the start of the R2 segment and _306_VQIVYK_311_ at the start of R3, are responsible for p-tau aggregation, as reported in the literature [26]. Since the most potent driver of tau aggregation is the VQIINK segment, we performed docking studies on the _275_VQIINK_280_ segment [27]. We selected the _274_KVQIINKKLD_283_ structure with PDB ID 5V5B from the Research Collaboratory for Structural Bioinformatics (RCSB) for performing molecular docking [28]. In the case of Aβ peptide aggregation, Aβ42 comparatively aggregates faster than other peptides. Because of that, we performed docking studies on the Aβ_1–43_ segment with PDB ID 2LLM. Based on ten different docking positions, the best poses that have the highest binding score were chosen. The docked compounds and their binding affinity in the ligand-receptor complexes are listed in Table 3. L_1–7_ show binding energies in the range of −4.6 to −5.6 kcal/mol toward p-Tau, and −4.9 to −6.0 kcal/mol for Aβ.

L_1_ showed the highest docking score towards p-tau and Aβ of all the investigated molecules. Among the seven molecules, two (L_1_ and L_3_) showed a higher binding score with p-tau than when compared to the reference molecule. Figure 3 represents the best binding poses of L_1–3_ with both proteins. L_1–3_ molecules showed a higher binding score (−5.3 to −6.0 kcal/mol) with Aβ than other inhibitor molecules, and L_2_ showed a binding score of −5.1 kcal/mol with p-tau, which very closely trails the reference compound. In further studies, we shortlisted three of these best-performing molecules (L_1–3_), since the present work aims to find out the lead inhibitor molecule. Figure 4 and Appendix A give a better insight into the interaction of the three selected molecules with Aβ and p-tau. The images have been generated with Pymol and the Discovery Studio visualizer program [29,30].

As seen in Figure 4, PHE8 is one of the most interactive amino-acid residues of Aβ that can bind through either π–π or π–alkyl interactions with selected inhibitor molecules. In L_1_, a hydrogen bond is formed between the oxygen atom in the molecule and LYS16 with a bond length of 2.20 Å, as listed in Appendix A. L_2_ shows the second-strongest binding affinity towards Aβ with a score of −5.8, which involves the following interactions: π–σ in LEU5, π–π stacked in PHE8, π–alkyl in PHE8, and alkyl in VAL12.

In the case of p-tau, ASN279, LYS280, LYS281, and LEU282 are the amino acids interacting with the inhibitor molecules. In the three selected inhibitor molecules, the most interactive amino-acid residues are LYS280 and LYS281. Apart from these, ASN279 and LEU282 are also participating in the molecular interactions. L_1_ is found to be stable at the active site of p-tau by forming π–alkyl (LYS280 and 281), π–σ (LYS 281), and alkyl (LYS280 and LEU282) interactions. The carbonyl oxygen and imine nitrogen of L_3_ formed a hydrogen bond with ASN279 and LYS281, respectively. Carbon–hydrogen bonding (ASN279), π–alkyl (LYS280 and LYS281), π–σ (LYS281), and alkyl (LEU282) are responsible for the interactions in L_3_.

### 2.4. Molecular Dynamics Simulations

Molecular dynamics (MD) simulation is used to evaluate the flexibility or mobility of different parts of the biomolecule and helps to predict how a biomolecular system will react to a perturbation [31]. The best docking poses were subjected to molecular dynamics simulation using the CHARMM36m force field, and the topology of the ligands was generated using the CHARMM-GUI platform [32,33]. The Monte Carlo method is used for ion placing, and molecules are equilibrated using the NVT and NPT isothermal–isobaric ensemble. Subsequently, we executed a 1 ns MD simulation with a 2 fs integrative step. The root-mean-square deviation (RMSD) was used to analyze the conformational stability of biomolecules. The overall stability and structural convergence of the protein–inhibitor complexes along with the apo-protein system were compared. Figure 5 represents the RMSD plot of Aβ’s and p-tau’s protein–inhibitor complexes and their apo forms. 

In the case of Aβ, apo and the inhibitor-bound forms deviate only to an acceptable extent, and the deviation range is only in the range of 0.1 to 1.5 nm. Among the selected three molecules (L_1–3_), the Aβ-L_1_ complex shows deviation from the apo-Aβ after 0.8 ns. The Aβ-L_3_ complex shows a slight variation from the apo-protein trend in the range of 0.2 to 0.5 ns. Apo-Aβ and the L_3_-bound forms show similar RMSD trends throughout the analysis. In the case of apo-p-tau, the RMSD plot of p-tau–L_1_ and p-tau–L_3_ showed a deviation in the range of 0.01 to 2.5 nm. The L_2_-bound p-tau showed a similar RMSD trend compared to others. After 0.7 ns, p-tau–L_2_ and p-tau–L_3_ deviate from the RMSD trend of the apo-p-tau. The L_3_ inhibitor molecule interacting with Aβ is in good agreement with its apo-protein RMSD trend. It also has a similar trend with apo-p-tau over the majority of the time-frame. The given RMSD data are acceptable because of a similar pattern of deviation observed even in the case of the apo-protein. The results show that L_2_ and L_3_ show similar RMSD deviation with apo-proteins with tau and Aβ, respectively. In the case of L_1_, both proteins show variation in the RMSD trends. On comparing L_2_ and L_3_ as dual inhibitors, L_3_ can be a better dual-inhibitor candidate compared to others. L_3_ shows a similar RMSD trend with apo-Aβ compared to the remaining molecules, while justifying the RMSD trend with p-tau deviates it to a certain extent.

### 2.5. TD-DFT Analysis and Solvent Effect

The TD-DFT method is used to determine the electronic properties of L_1_, L_2_, and L_3_ in their excited states. The wavelength (λ), the excitation energy, the electronic oscillator strength (f), and the corresponding electronic transition in the gaseous phase are all provided in Appendix A. The optimized geometries of L_1_, L_2_, and L_3_ revealed an intense absorption peak at 291, 299, and 249 nm, respectively. Regarding this absorption peak, L_1_ showed maximum contribution from the transition HOMO-1→LUMO+1 (40%) at 291 nm with f = 0.3055. In L_2_, the transitions are mainly attributed to the HOMO-1→LUMO+1 (46%), HOMO-1→LUMO+2 (13%), and HOMO-5→LUMO (13%), HOMO→LUMO (11%), respectively. In the case of L_3_, HOMO-4→LUMO+1, HOMO-4→LUMO, and HOMO-1→LUMO+2 transitions contribute 22%, 10%, and 8%, respectively. A value of f close to one denotes an allowed transition, whereas an f close to 0.001 denotes a quantum mechanically forbidden transition. The selected molecules show an f value between 0.2 and 0.3, which denotes allowed transitions. The optimized geometries in the water medium of L_1_, L_2_, and L_3_ revealed strong absorption peaks at 315, 333, and 232 nm, respectively (For detailed information, see the Appendix A). L_2_ showed the highest absorption peak at 333 nm with maximum contribution from HOMO-1→LUMO (63%). In the case of L_1_ and L_2_, the wavelength of the absorption peak is increased in the water medium compared to the gaseous phase. L_3_ had an intense absorption peak at 232 nm, with f = 0.4720. The f value of all three inhibitor molecules in the water medium is increased compared to the gaseous phase. The obtained electronic oscillator strength values indicate that the molecules can absorb light more efficiently.

The molecular electrostatic potential (MEP) map, which is based on electronic density and chemical reactivity sites, provides details on the total charge distribution of the molecule. The reactive areas of the molecule can therefore be easily determined using the MEP surface. The MEP map given in Figure 6 shows the colors red, orange, yellow, green, and blue, which are related to the distribution of the electronic densities. The difference in MEP in both the gaseous and solvent (water) phases will provide a quantitative perspective of the molecular characteristics under physiological conditions. In L_1_ and L_2_, the NO_2_^−^ group attached to the phenyl ring is distributed over the red region in the gaseous phase and the dark-red region in the water medium. This indicates that, in the water medium, molecules become more nucleophilic and their interactions with the protein may be increased. In L_2_ and L_3_, the pyrrolidine ring becomes slightly electron deficient in a water medium. The carbonyl group of L_3_ in the water medium becomes more nucleophilic than in the gaseous phase. The oxygen of the carbonyl group has an interaction with the ASN279 of tau with a bond length of 2.5 Å, which may be even stronger in the water medium because of the increase in nucleophilicity. The comparison of band-gap energy values of the molecules in the gaseous and solvent phases is listed in Appendix A. The band gap of all three molecules is slightly decreased in the water phase compared to the gaseous phase. Therefore, the solvent phase studies indicate the reactivity of molecules increased under a water medium that simulates physiological conditions. This means the probability of inhibitor–protein interactions is higher in physiological conditions.

## 3. Materials and Methods

The absorption, distribution, metabolism, excretion, and toxicity (ADMET) properties of the compounds were predicted by using the SwissADME and ProTox-II tools [34,35,36,37]. Target prediction for the inhibitor molecules was recorded with the SwissTargetPrediction. Intestinal absorption and blood–brain barrier (BBB) penetration properties were determined by the BOILED-Egg method [38]. The geometries of the molecules considered in this work have been initially optimized using density functional theory (DFT) at the ωB97XD/6-311++G(2d, 2p) [39,40,41] level of theory. We used (E)-configuration structures in all the computational calculations throughout this work, which is the minimum-energy structure in the potential energy surface. This fact is well supported in the literature [16,42]. The time-dependent density-functional theory (TD-DFT), frontier molecular orbital (FMO), and molecular electrostatic potential (MEP) surface analyses are performed at the same level of theory. The protein-ligand docking process is carried out using AutoDock Vina [43,44]. The binding mechanism between the molecules and their corresponding proteins is examined using molecular dynamics simulation, which is carried out using NAMD/VMD [45,46]. The electronic transitions of the compounds are calculated by using the TD-DFT method [47,48]. All the geometry optimization calculations are performed with the Gaussian suite of programs [49].

## 4. Conclusions

In summary, we report the results of the in silico analysis performed on the potential dual Aβ and p-tau inhibitor molecules for AD treatment. We designed seven synthetically possible inhibitor molecules (L_1–7_) as dual Aβ and p-tau inhibitors. The binding affinity of these molecules towards both Aβ and p-tau is confirmed by docking analysis, which agrees well with the early pharmacodynamic assessments. All seven molecules exhibited BBB-crossing and gastrointestinal-absorption ability, which is essential for AD treatment. Out of these seven molecules, L_1–3_ were shortlisted for further investigation due to their higher docking score towards both proteins. LYS280 and LYS281 are the most interacting amino-acid residues in p-tau, and PHE8 is the most interactive amino-acid residue of Aβ. As per the MD simulation data, L_3_ possesses the highest similarity with the apo-proteins. Hence, L_3_ could be a lead molecule out of the seven investigated systems. L_1_ and L_2_ showed variance in the RMSD analysis from apo-Aβ. Therefore, L_1_ and L_2_ are potential molecules only for single-target inhibition of p-tau aggregation. Although the results point to L_3_ as the lead molecule, in-depth experimental studies are required to confirm the efficiency of all the inhibitor molecules (L_1–7_). Hence, these molecules can be considered for experimental studies, as they fulfilled the prerequisites of being a drug candidate when assessed theoretically. We put forward these molecules in the experimental analysis, since they can be prepared through different synthetic routes. The results show that the molecules have outstanding pharmacological characteristics with remarkable potential as AD-treating drugs. Further, this work will enlighten theoretical and experimental researchers involved in developing novel pharmacophores. This study can further act as a stepping stone for the auxiliary development of small organic multi-target inhibitor molecules in AD treatment.

## Figures and Tables

**Figure 1 molecules-28-01388-f001:**
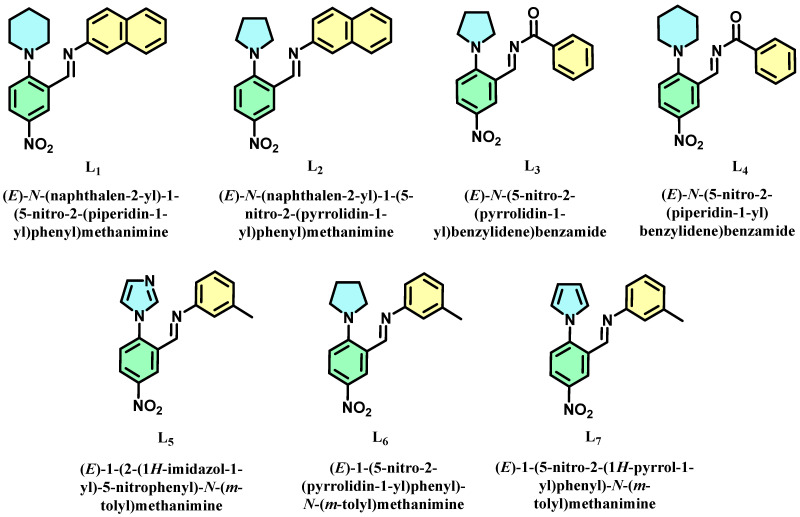
Molecular structures of seven dual inhibitors (L_1–7_).

**Figure 2 molecules-28-01388-f002:**
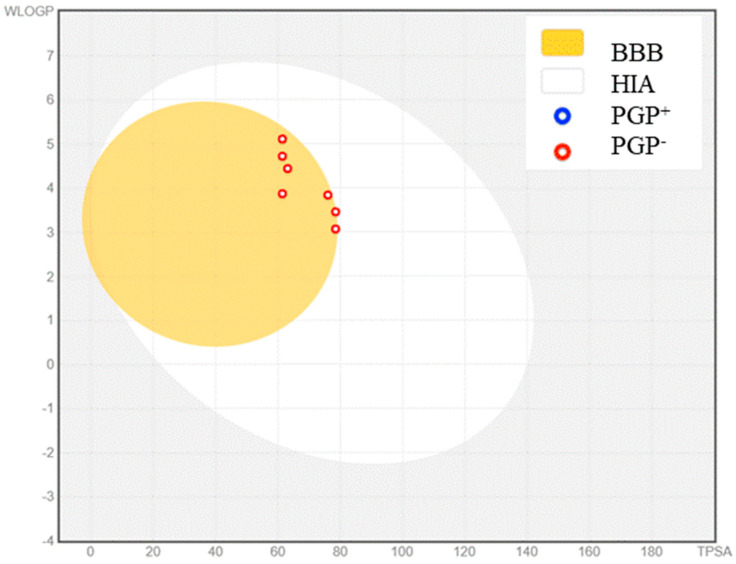
BOILED-Egg model of L_1–7_ to predict gastrointestinal absorption and BBB penetration. (HIA—human gastrointestinal absorption; PGP^+^—substrate of P-glycoprotein; PGP^−^—non-substrate of P-glycoprotein).

**Figure 3 molecules-28-01388-f003:**
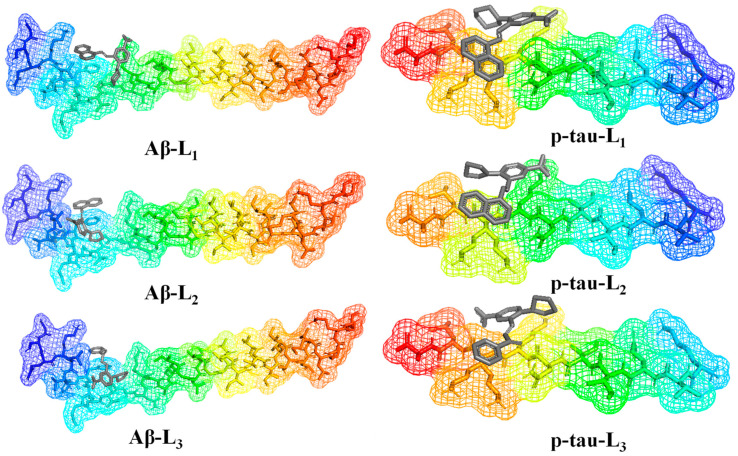
Docking poses of L_1–3_ in the binding site of Aβ and p-tau.

**Figure 4 molecules-28-01388-f004:**
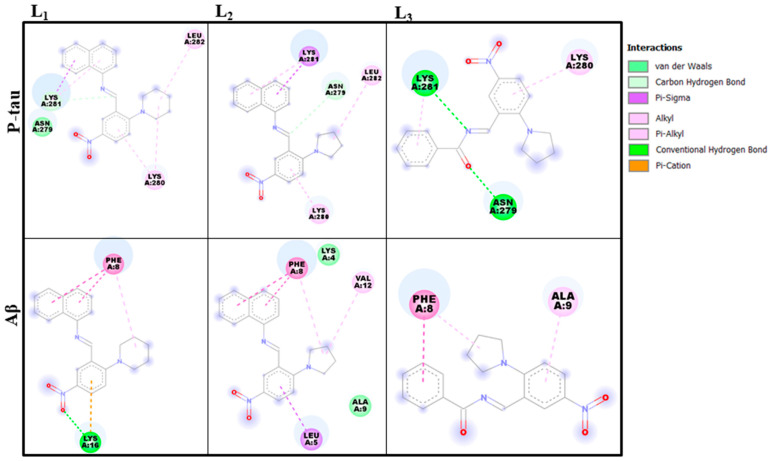
Two-dimensional representation of interactions between L_1–3_ and proteins.

**Figure 5 molecules-28-01388-f005:**
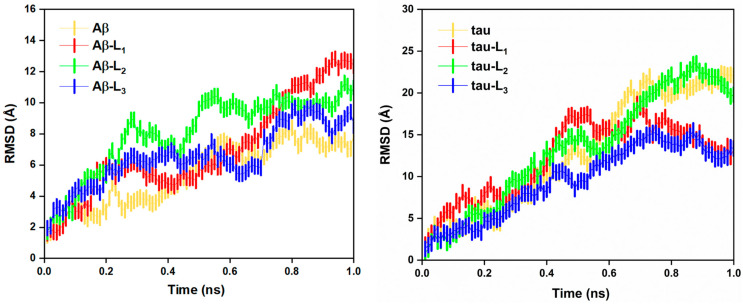
Root-mean-square deviation (RMSD) plots of Aβ and p-tau in both apo (in yellow) and with inhibitor-bound forms (L_1_, L_2_, and L_3_ are in red, green, and blue respectively) in the molecular simulation of 1 ns.

**Figure 6 molecules-28-01388-f006:**
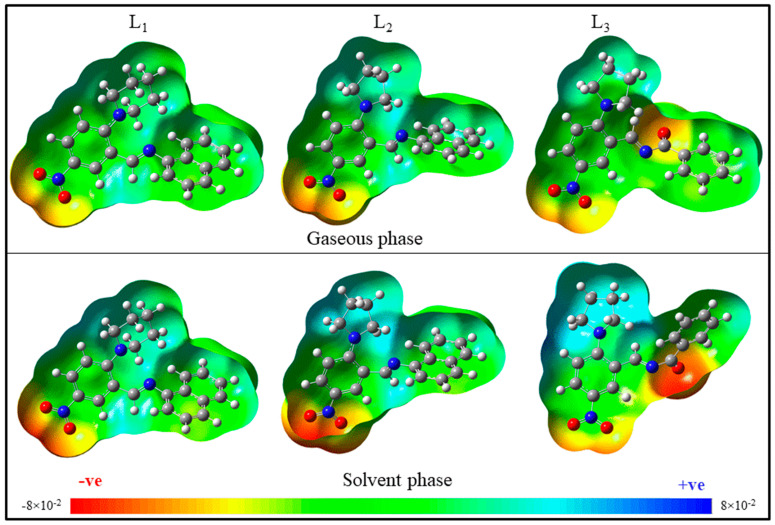
Molecular electrostatic potential map of L_1–3_.

**Table 1 molecules-28-01388-t001:** In silico drug-likeness properties of L_1–7_ obtained using the SwissADME tool.

Molecules	MW	TPSA	Log P	ESOL Log S	Log K_p_ (cm s^−1^)	Violations		Synthetic Accessibility	Target	
						Lipinski	Ghose		MAPT	Amyloid Beta
L_1_	359.42	61.42	4.15	−5.56	−4.75	0	0	3.05	yes	yes
L_2_	345.39	61.42	3.88	−5.27	−4.92	0	0	2.99	yes	yes
L_3_	323.35	78.49	2.91	−4.11	−5.76	0	0	2.89	yes	yes
L_4_	337.37	78.49	3.11	−4.41	−5.59	0	0	2.96	yes	yes
L_5_	306.32	76	2.51	−4.02	−5.92	0	0	2.83	yes	yes
L_6_	309.36	61.42	3.15	−4.42	−5.33	0	0	2.89	yes	yes
L_7_	305.33	63.11	3.11	−4.42	−5.46	0	0	2.76	yes	yes

MW—molecular weight; TPSA—topological polar surface area; log P—lipophilicity of arithmetic mean of the values predicted by the five proposed methods; ESOL log S—water solubility by estimated solubility method; log K_p_—permeability coefficient; MAPT—microtubule-associated protein tau.

**Table 2 molecules-28-01388-t002:** The computed global reactivity descriptors for L_1–7_.

Parameters/Descriptors	L_1_	L_2_	L_3_	L_4_	L_5_	L_6_	L_7_
E_HOMO_ (eV)	−7.73	−7.62	−8.43	−8.12	−8.30	−7.77	−8.06
E_LUMO_ (eV)	−0.74	−0.49	−0.86	−0.62	−1.31	−0.43	−1.13
Energy band gap (eV) (ΔE = E_LUMO_ − E_HOMO_)	6.99	7.13	7.57	7.50	6.99	7.34	6.93
Ionization potential (I = −E_HOMO_) (eV)	7.73	7.62	8.43	8.12	8.30	7.77	8.06
Electron affinity (eV) (A = −E_LUMO_)	0.73	0.48	0.85	0.62	1.30	0.42	1.12
Chemical hardness (η = (I − A)/2)	3.49	3.56	3.78	3.75	3.49	3.67	3.46
Chemical softness (σ = 1/2 η)	0.14	0.14	0.13	0.13	0.14	0.13	0.14
Electronegativity (χ = (I + A)/2)	4.23	4.05	4.64	4.37	4.80	4.10	4.59
Chemical potential (μ = −(I + A)/2)	−4.23	−4.05	−4.64	−4.37	−4.80	−4.10	−4.59
Electrophilicity index (ω = μ^2^/2η)	2.56	2.30	2.84	2.55	3.30	2.29	3.04
Maximum charge transfer index (ΔN_max_ = −μ/η)	1.21	1.13	1.22	1.16	1.37	1.11	1.32

**Table 3 molecules-28-01388-t003:** The molecular docking scores of L_1–7_ with Aβ and p-tau.

Molecules	Docking Score (kcal/mol)	
	Amyloid Beta	p-tau
L_1_	−6.0	−5.6
L_2_	−5.8	−5.1
L_3_	−5.3	−5.4
L_4_	−5.2	−4.6
L_5_	−5.0	−5.0
L_6_	−5.0	−5.0
L_7_	−4.9	−5.2
Reference molecule [16]	-	−5.2

## Data Availability

The data presented in this study are available in the article and Appendix A also available.

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
