# Peer review of "In Silico Drug Design and Analysis of Dual Amyloid-Beta and Tau Protein-Aggregation Inhibitors for Alzheimer’s Disease Treatment"

_molecules, 2023, doi:10.3390/molecules28031388_

Round 1

Reviewer 1 Report

The manuscript presents an interesting and important In silico drug design and analysis of dual Amyloid beta and tau 2 protein aggregation inhibitors for Alzheimer's disease treatment.

Specific comments on the manuscript:

1) The aim of the study is not clearly defined. The content of some parts of the work is not clear and the study design should be corrected.

2) You should emphasize the importance of this study and its novelty.

3) The combination of results and discussion sections is not acceptable. In the present form the manuscript should be rejected.

4) The conclusion should be more informative. The authors should link what has been found in the study and what could be the implications for practice.

Author Response

We have uploaded the file for the reviewers to reply

Reviewer 2 Report

The manuscript describes a computational study combining DFT, docking and MD simulations to evaluate suitability of piperidine-substituted 3-nitrobenzaldehyde analogues as dual inhibitors of A-beta and p-tau aggregation. The analysis also has been extended to evaluate ADMET properties, BBB properties, frontier molecular orbital as well as molecular electrostatic potential.

Following needs to be addressed before the manuscript could be accepted for publication.

(i)             Authors describe that ‘….from the obtained results, we short-listed seven dual inhibitor molecules …’. The details of the larger dataset considered is needed and what was the criteria used to select these 7 compounds?

(ii)            All these molecules are likely to exist in at least 2 unique conformations, defined by the rotation around nitrobenzene ring and methanimine group. Compounds L1 and L2 are shown in one possible conformation while compounds L3-L7 are shown in the alternate conformation. These different conformations are clearly visible in Fig 6 as well. What was the basis for choosing these conformations? It would be appropriate to generate alternate conformations for each compound.

(iii)           DFT and TD-DFT calculations and subsequent FMO and MEP analysis would be very sensitive to geometry (conformation) of the molecule. More details are need on how these conformations were chosen.

(iv)           MD simulations has been described using a single line employing NAMD/VMD. What force field was used to describe protein and ligand molecules? Additional details of the parameters used for MD simulations are needed. 100 ps is very short MD simulation that would not even qualify as an equilibration of the system. The large fluctuations and deviations in RMSD indicated that none of the systems are stable in MD simulations and/or converged within 100 ps. MD simulations need to be extended and performed in duplicate/triplicate (starting from different structure/conformation or initial velocity distribution) to obtain statistically significant results.

Author Response

(The authors gave the same response as above.)
